# Wine’s Phenolic Compounds and Health: A Pythagorean View

**DOI:** 10.3390/molecules25184105

**Published:** 2020-09-08

**Authors:** Francesco Visioli, Stefan-Alexandru Panaite, Joao Tomé-Carneiro

**Affiliations:** 1Department of Molecular Medicine, University of Padova, Viale G. Colombo 3, 35121 Padova, Italy; stefan.panaite@outlook.it; 2IMDEA-Food, CEI UAM + CSIC, 28049 Madrid, Spain; joao.estevao@imdea.org

**Keywords:** wine, polyphenols, flavonoids, diet, clinical trials, metabolites

## Abstract

In support of the J curve that describes the association between wine consumption and all-cause mortality, researchers and the lay press often advocate the health benefits of (poly)phenol consumption via red wine intake and cite the vast amount of in vitro literature that would corroborate the hypothesis. Other researchers dismiss such evidence and call for total abstention. In this review, we take a skeptical, Pythagorean stance and we critically try to move the debate forward by pointing the readers to the many pitfalls of red wine (poly)phenol research, which we arbitrarily treat as if they were pharmacological agents. We conclude that, after 30 years of dedicated research and despite the considerable expenditure, we still lack solid, “pharmacological”, human evidence to confirm wine (poly)phenols’ biological actions. Future research will eventually clarify their activities and will back the current recommendations of responsibly drinking moderate amounts of wine with meals.

## 1. Introduction

The association between alcohol consumption and health follows a J-shaped curve [1]. Moderate alcohol use is associated with better prognosis and lower all-cause death, whereas excessive intake is detrimental to human health [1]. The mechanisms underlying the protective effects of moderate alcohol consumption are under investigation and mostly involve reduced plasminogen levels and lower thrombogenicity observed in moderate drinkers vs. abstainers [1]. Some authors propose the superiority of wine, namely red wine over other alcoholic beverages and attribute such advantage to the (poly)phenolic components of red wine [2]. Even though this notion is not fully proven and is, conversely, often challenged [2], much research is being performed to elucidate the purported biochemical mechanisms through which wine (poly)phenols would afford better health, in particular by lowering cardiovascular risk. The debate on alcohol use and health is becoming heavily polarized: one party underscores a large amount of data in support of the J curve [1,3] whereas the other side dismisses such evidence and calls for total abstention [2,3]. In support of the former, researchers and the lay press often advocate the health benefits of (poly)phenol consumption via red wine intake and cite the vast amount of in vitro literature that would corroborate the hypothesis [4,5].

In this review, we take a skeptical, Pythagorean stance and we critically try to move the debate forward by pointing the readers to the many pitfalls of red wine (poly)phenol research, which we arbitrarily treat as if they were pharmacological agents.

## 2. Phenolic Compounds and Health

Diet and nutrition are essential to promote and maintain good health throughout life and for many years they have been known to be of crucial importance as risk factors for chronic diseases, making them essential components of prevention activities [6]. The consumption of foods derived from plant products such as wine, fruits, vegetables, nuts, cereals, legumes, spices and others integrated into the Mediterranean or the DASH diets [7], is associated with beneficial health effects and a protective role against the development and progression of diseases, such as cardiovascular disease (CVD) [8]. The ability of some plant-derived foods to reduce disease risk has been associated with the presence of non-nutrient secondary metabolites (phytochemicals) to which a wide variety of biological activities are attributed [9,10]. These metabolites have moderate potency as bioactive compounds and low bioavailability compared to drugs, but when ingested regularly and in significant amounts can have noticeable mid/long term physiological effects. Phytochemicals present in foods associated with a beneficial health effect include glucosinolates, terpenoids and a large group of phenolic compounds (anthocyanins, flavones, flavan-3-ol, stilbenes, etc.) collectively known as (poly)phenols [10,11].

## 3. Classification and Amounts of Wine Phenolic Compounds

Phenolic compounds have as a common characteristic in their chemical structure the presence of one or more hydroxyl groups attached to one or more aromatic or benzene rings. In general, phenolic compounds that contain more than one phenolic group are called polyphenols to distinguish them from simple phenolics. Typically, these compounds are found in a conjugate form with one or more sugar residues linked by β-glycosidic (*O*-glycosylated) bonds or by direct linkages of sugar to an aromatic ring carbon atom (C-glycosides) [12]. Phenolic compounds are grouped according to their chemical structure into two main categories, flavonoids and non-flavonoids, each comprising several sub-groups. In wine, sub-groups of flavonoids compounds include flavonols, flavononols (also known as dihydroflavonols), anthocyanins, flavan-3-ols, flavanones and flavones, while non-flavonoids contain hydroxycinnamic and hydroxybenzoic acids, and stilbenes (Table 1). (Poly)phenolic composition varies among different wines according to the type of grape used, vinification process used, type of yeast that participates in the fermentation, and whether grape solids are present in the maceration process [13]. For instance, in grapes, the composition in phenolic compounds is location-dependent, i.e., pulp, skin and seeds have different types and proportions of (poly)phenols); since red wines are exposed to all grape parts during the vinification process they contain more (poly)phenols than white wines, whose contents essentially originate from the pulp. In this sense, the minimum and maximum levels of total phenolic contents reported in a representative set of studies (expressed as the median (Q25–Q75) in mg of gallic acid equivalents (GAE) per liter) were 1531 (983–1898) and 3192 (2700–3624), and 210 (89–282) and 402 (347–434) for red and white wines, respectively (Table 2). The content of polyphenols in rosé wine is intermediate between red and white wines [14,15].

### 3.1. Flavonoids

Flavonoids have a skeleton with 15 carbon atoms and are represented in a C6-C3-C6 type system, where a benzene ring (designated as B) is joined (in most cases) to the C2 position of a γ-pyran type ring (C) included in a chromane ring (Table 1) [33]. The structure of flavonoids is shaped by different levels of hydroxylation, prenylation, alkalization or glycosylation reactions, which give rise to different sub-groups [34]. In plants, most flavonoids exist as glycosides in combination with monosaccharides such as glucose and rhamnose (most common), followed by galactose, xylose and arabinose [35].

### 3.2. Flavonols

Flavonols are characterized by a hydroxyl group in C3 (Table 1) and are often named 3-hydroxyflavones. These compounds are known to play a wide range of biological activities and are considered the main active compounds within the flavonoids group [36,37]. Flavonols and their glycosides are present in red and in white wines, influencing their color, taste, and health properties [38]. Flavonols in red wine include aglycons such as myricetin, quercetin, kaempferol, and rutin and their respective glycosides (glucosides, glucuronides, galactosides and diglycosides). Quercetin 3-*O*-glucoside is the most representative flavonol in wines [39]. Flavonol levels in red wine can reach over 150 mg/L (Table 3).

### 3.3. Flavan-3-Ols 

Flavanols or flavan-3-ols are responsible for the astringency, bitterness, and structure of wines, and are found in important concentrations in red wine [40]. They are benzopyrans, having no double bond between C2 and C3, and no C4 carbonyl in Ring C. Furthermore, due to the hydroxylation at C3 flavanols have two chiral centers. (+)-Catechin (*trans* configuration) and (−)-epicatechin (*cis* configuration) are the two main flavan-3-ol isomers found in red wine, with average combined concentration over 100 mg/L (Table 3). Catechins usually occur as aglycones, or esterified with gallic acid, and can form polymers, which are often referred to as proanthocyanidins (or condensed tannins) because an acid-catalyzed cleavage of the polymeric chains produces anthocyanidins. Proanthocyanidins, which present an average concentration over 350 mg/L in red wine, include, for example, procyanidin dimers B1, B2, B3 and B4. Trimers such as procyanidin C1 (three epicatechins) have also been identified.

## 4. Anthocyanins

Anthocyanic pigments (anthocyanidins and anthocyanins) have a structure based on the flavylium cation (2-phenylbenzopyrylium). In fact, anthocyanins are anthocyanidin glycosides, being the corresponding aglycons (anthocyanidins) obtained by hydrolysis. The great variety of anthocyanins found in nature (more than 500 anthocyanins have been described) is characterized by the different hydroxylated groups, conjugated sugars and acyl moieties they present [41,42]. The main anthocyanidins found in red wine are malvidin (most abundant), petunidin, peonidin, delphinidin and cyanidin. Often anthocyanins are found linked (mainly in position 3) to one or more sugar molecules, usually glucose, and also to acyl substituents bound to sugars, aliphatic acids, and cinnamic acids. Anthocyanins can be present in amounts higher than 700 mg/L in red wine, whereas in white wine they are essentially absent (Table 3).

### 4.1. Flavanones

Flavanones, also known as dihydroflavones, lack the double bond between carbons 2 and 3 in the C-ring of the flavonoid skeleton. Some flavanones have unique substitution patterns, e.g., prenylated flavanones, furanoflavanones, pyranoflavanones, benzylated flavanones, resulting in a large number of derivatives of this subgroup. One of the main flavonones found in wine is naringenin at levels that can reach 25 mg/kg (Table 3).

### 4.2. Flavones

Flavones are characterized by absence of a hydroxyl group in the C3 position and a conjugated double bond between C2 and C3 in the flavonoid skeleton. Flavones and their 3-hydroxy derivatives flavonols, including their glycosides, methoxides and other acylated products on all three rings, make this the largest subgroup among all polyphenols. These compounds were found in grape skin and wine in both aglycones and glycosides forms. Apigenin, for example, has been described in red wine only in trace amounts (Table 3).

## 5. Non-Flavonoids

The non-flavonoid phenolic constituents in wine are divided into hydroxybenzoic acids and hydroxycinnamic acids, stilbenes and other miscellaneous compounds [43]. These phenolic compounds can reach levels that range from 60 to 566 mg/L [44].

## 6. Hydroxycinnamic Acids

Hydroxycinnamic acids are the foremost group of phenolic compounds in grapes and wine [45]. Caffeic, coumaric, and ferulic acids, essentially conjugated with tartaric acid esters or diesters, are some of the most important compounds in this polyphenol sub-class. For instance, caftaric acid, which is composed of caffeic acid esterified with tartaric acid, is found in the pulp and represents up to 50% of total hydroxycinnamic acids [43,46]. The average amount of hydroxycinnamic acids present in red wine is around 100 and 30 mg/L in red and white wines, respectively (Table 3).

## 7. Hydroxybenzoic Acids

In comparison with cinnamic acid derivatives, benzoates are present at lower levels in wine (Table 3). Hydroxybenzoic acids are phenolic metabolites with a general C6–C1 structure and occur mainly in their free forms in wine, mainly as *p*-hydroxybenzoic, gallic, vanillic, gentisic, syringic, salicylic, and protocatechuic acids [43], although ethyl and mehyl esters of these phenolic acids have been also identified [47]. Gallic acid, which is present in important levels in white and, especially, in red wine, is the precursor of all hydrolyzable tannins and is incorporated in condensed tannins [46].

## 8. Stilbenes

Stilbenes are widely distributed molecules in the Plant Kingdom. However, their presence in the diet is not very significant, being basically restricted to grapes, red wine and, to a lesser extent, peanuts and blueberries [48]. Chemically they are 1,2-diarilethenes and usually have two hydroxyl groups in the meta position of ring A, while ring B is substituted with hydroxyl groups and methoxyl groups in the meta and/or para positions (Table 1). Although its concentration in wine is much lower than other polyphenols, i.e., often traces, resveratrol has received much attention for its biological properties and potential therapeutic effects (see below). The levels of resveratrol aglycone, its piceid glycoside, and its dimeric and trimeric forms (e.g., pallidol, viniferins) combined may range from negligible up to more than 100 mg/L (Table 3) when grapes are exposed to fungi.

## 9. Effects on Human Health

The Greek philosopher Pythagoras of Samos allegedly used to say “All is number” or “God is number” [49]. He meant that he only believed in what could be measured. This was echoed by William Thomson, 1st Baron Kelvin who, in his *Popular Lectures and Addresses* vol. 1 (1889) ‘Electrical Units of Measurement’, delivered 3 May 1883 notoriously said “When you can measure what you are speaking about, and express it in numbers, you know something about it, when you cannot express it in numbers, your knowledge is of a meager and unsatisfactory kind; it may be the beginning of knowledge, but you have scarcely, in your thoughts advanced to the stage of science.” [50]. What both scientists meant was that we should base our knowledge on hard evidence. More recently, Dr. Archie L. Cochrane set out clearly the vital importance of randomized controlled trials (RCTs) in assessing the effectiveness of treatments [51]. How does this apply to wine (poly)phenols?

We shall start by mentioning that there are thousands of papers published on this topic (a cursory PubMed search ran on August 14th, 2020 retrieved 2954 entries just by entering “wine polyphenols”). The near totality of such studies has been performed in vitro. Needless to say, in vitro studies are indispensable to address mechanisms of action and to propose new avenues of in vivo research. The case of wine (poly)phenols, however, is rather unique and presents us with a paradigmatic opportunity to underscore the current very limits of (poly)phenol research.

In keeping with the above, we would like to discuss the case of resveratrol as an example of molecules for which there exists a strong dyscrasia between the lay public perception of health benefits and hard scientific data.

Resveratrol became popular in 1991, when Drs. Michel de Lorgeril and Serge Renaud appeared in the “60 Minutes” CBS show to talk about the French Paradox and to attribute it to the French habit of drinking red wine, which would theoretically inhibit lipid peroxidation. Note that, back then the “free radical/antioxidant hypothesis” [52] was in full swing and it was commonplace to believe that eating and drinking (poly)phenols would scavenge free radicals and prevent their noxious effects, for example by inhibiting LDL oxidation [53]. This conjecture, now largely proven wrong [54], granted red wine (poly)phenols, namely resveratrol, immediate popularity and trigger the vast amount of well-funded research mentioned above.

Two major issues developed during the nearly three decades that separate the 60 min show from our current knowledge.

The first problem is that we came to realize that (poly)phenols are very weak (if at all effective) in vivo direct antioxidants [55]. For kinetic reasons they do not scavenge free radicals and their bioavailability is generally so low that they contribute very little to the integrated cellular antioxidant machinery, which is mostly composed of enzymes [56,57,58]. Alas, plenty of investigators still perform research and publish papers on the in vitro antioxidant abilities of individual (poly)phenols or of some raw mixtures of them. Luckily, plenty of researchers correctly use (poly)phenols’ metabolites in their in vitro studies [59,60]. The hurdle then becomes the difficulty of synthesizing such metabolites, which are often produced by the organism in different forms. It is worth underscoring that we are making progress in the identification of metabolites, but—until recently—we mainly focused on the liver-derived ones [61]. The relatively recent discovery of microbiota-synthesized metabolites amplifies the list of potential biologically-active molecules produced by the body after the ingestion of (poly)phenol-rich foods [58,62,63].

In consonance with the above, the lay press often overlooks the bioavailability issue. As regards resveratrol, already in 1993 Soleas and Goldberg acted as the harbinger of the subsequent in vivo debacle of the molecule by calling it “a molecule whose time has come and gone” [64]. That conclusive title might have been a bit too harsh, but it’s a fact that many years of research and many million dollars invested in it did not yield major results [65,66].

Finally, animal studies often employ very high doses of grape (poly)phenols, e.g., resveratrol and their results cannot be readily transferred to humans, who would need to ingest several grams of extracts to replicate the same effects. Indeed, a discrepancy between animal and human effects has just been underscored [5] and resveratrol’s potential toxicity has been recently reviewed [67]. An often overlooked paper reported that resveratrol promoted atherosclerotic development in hypercholesterolemic rabbits, by a mechanism that is independent of observed differences in gross animal health, liver function, plasma cholesterol concentrations, or LDL oxidative status [68].

## 10. Human Studies of Resveratrol and Red Wine (Poly)Phenols

One of the fields where red wine (poly)phenols are most actively studied is that of weight control, namely obesity and its associated insulin sensitivity [69]. The rationale behind studying red wine (poly)phenols and, particularly, resveratrol is that type II diabetes is rampant in developed countries and that many researchers are looking for fasting mimetics, to approximate the beneficial effects of calorie restriction or intermittent fasting on insulin sensitivity [70]. The results are equivocal, to say the least, as most trials failed to report significant effects, e.g., [71]. The molecular rationale for studying it is the finding that resveratrol and, maybe, other wine (poly)phenols activate SIRT1, a modulator of pathways downstream of calorie restriction that produces beneficial effects on glucose homeostasis and insulin sensitivity [72,73]. This hypothesis is quite controversial for at least two reasons. One is the factual role of sirtuins as longevity promoters [74]. The other one is that several researchers question the reproducibility of those data, e.g., [75]. In summary, the jury is still out [76] and the quixotic search for a substance that would fix the cardiometabolic effects of inordinate diets is not over [77].

Rather than trying to single out individual molecules purportedly responsible for the beneficial effects of moderate wine use (which is a pharmacological approach), an alternative is to test the effects of the whole (poly)phenolic fraction. We retrieved 24 publications of human studies that employed dealcoholized wine (Appendix A). Taken together, their results indicate that wine (poly)phenols do exert healthful effects, ranging from anti-inflammatory actions to modulation of the microbiota, which is now gaining traction from an industrial viewpoint [78,79] and might be one of the next applications of these compounds. The extent and precise nature of such activities, however, remains to be fully elucidated. For example, some publications stem from the same study; there are some contradictions between data and their discussion (e.g., LPS and LPB data in [80], fatty acid data in [81], inflammatory markers in [82], etc.); and the true clinical relevance of microbiota modification as related to, e.g., circulating lipids (Appendix A) [83,84,85,86,87,88,89,90,91,92,93,94,95,96,97,98,99,100,101,102,103,104,105,106]. In summary, there is indeed evidence that wine (poly)phenols modulate human physiology, but puffery should be avoided until we can clearly correlate such modifications with undisputable clinical outcomes.

We also searched the literature for acute or short-term human effects of wine (poly)phenols (Appendix A) [107,108,109,110,111,112,113,114,115,116,117,118,119,120,121,122,123]. Even though this might be seen as a more classic “pharmacological” approach, even small effects repeated over time might—in the end—affect human physiology and health. Some outcomes fall in the now-outdated “plasma antioxidant capacity” or “oxLDL” areas, i.e., are poor proxies of prognosis. Other data are more physiologically relevant and indicate, e.g., salubrious effects on endothelial function and related flow-mediated dilation. Anti-inflammatory effects have also been reported. Other studies focused on bioavailability, with scant indications of biological effects. It is worth noting that ethical reasons often impede research on alcohol in humans [1].

## 11. Wine vs. Other Alcoholic Beverages: Does Digestion Make the Difference?

Often miscategorized as direct antioxidants [124] (see above) wine (poly)phenols might act as such during digestion. Several pieces of evidence reveal that, during digestion, lipid peroxides are formed in the stomach at millimolar concentrations [125]. In addition, we eat pre-formed hydroperoxides, whose formation is unavoidable in fat-containing foods. Dr. Kanner called the stomach “a bioreactor” [125] where hydroperoxides are formed and subsequently absorbed. This is particularly noteworthy in the case of red meat (hypothetically because of its iron content [126]), but it is likely to happen with any animal food. Lipid peroxidation during digestion can be decreased by the consumption of (poly)phenol rich foods and beverages such as extra virgin olive oil [127] and—germane to this review—red wine [128,129]. In a wider context, these data experimentally explain the evolutionary-sound habit of eating fruit and vegetables, i.e., (poly)phenols with protein [130]. Further, most cultures have culinary routines that involve drinking (poly)phenols during or after meals, including tea [131], coffee [132,133], red wine [128], etc.

Another place where wine (poly)phenols might act as indirect antioxidants is the liver, where ethanol is metabolized to acetaldehyde by the microsomal ethanol oxidizing system (MEOS), via cyp2E1 [134]. In doing, ROS are generated as by-products. Possibly, (poly)phenols might lessen this untoward effect of ethanol ingestion, through mechanisms that are yet to be elucidated.

## 12. Conclusions

In this review, we took a pharma-nutritional approach to wine (poly)phenols. Epidemiological evidence describes the association between alcohol use and all-cause mortality as following a J-shaped curve. Many investigators claim the superiority of wine, namely red wine with respect to other alcoholic beverages and call for (poly)phenols to support their hypothesis. The result is that the lay public often believes in this premise, in part because there is plenty of in vitro and some animal data and in part due to wish bias [135]. Indeed, there appears to be a discrepancy between the strength of biochemical data and the scantiness of well-controlled human trials of individual molecules isolated from wine. This is commonplace in nutritional research [136,137] and there is no easy way out of it [138]. In a way, wine (poly)phenols are paradigmatic of the current tension between treating such compounds as non-essential nutritional agents and expecting pharmacological actions from them [139]. In the former case, we must accept the fact that the biological effects of wine (poly)phenols are minimal and very difficult to detect with current technologies and biomarkers [138,139]. The latter scenario involves the unavoidable acceptance of side effects and is not epistemologically applicable to human nutrition. Another limitation of wine (poly)phenol research is that we often use a reductionist approach and look for one single mechanism of action. In the case of wine (poly)phenols and particularly resveratrol, this involves their misclassification as in vivo free radical scavengers and antioxidants, even if their mechanisms of action are manifold and chiefly involve anti-inflammatory actions and, possibly, activation of nrf2 and its downstream pathways via xeno-hormesis [1]. In pharma-nutritional research we should look at a wider picture and acknowledge that phytochemicals contribute to health even though, based on the definition of nutrients, they are not essential. Therefore, these molecules do not fit in the classic and rigorous pharmacological definitions; they can be modified by organisms before they interact with targets, can have different targets depending on their concentration, and do not have a univocal pharmacological mechanism of action.

In conclusion, after 30 years of dedicated research and despite the considerable expenditure, we still lack solid, “pharmacological” human evidence to confirm wine (poly)phenols’ biological actions (Figure 1). Future research [138] will eventually clarify their activities and will back the current recommendations of responsibly drinking moderate amounts of wine with meals.

## Figures and Tables

**Figure 1 molecules-25-04105-f001:**
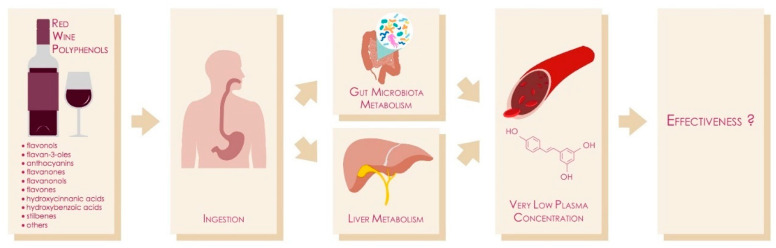
Schematic overview of current pitfalls in wine (poly)phenols research.

**Table 1 molecules-25-04105-t001:** Classification of phenolic compounds found in wine.

Group	Subgroup	Main Parent Compounds and Representative Derivatives
**Flavonoids** 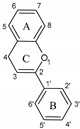	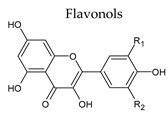	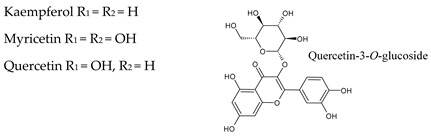
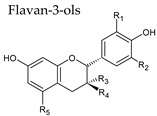	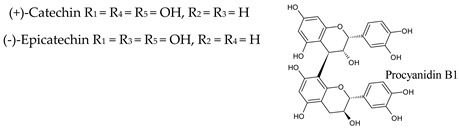
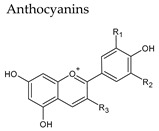	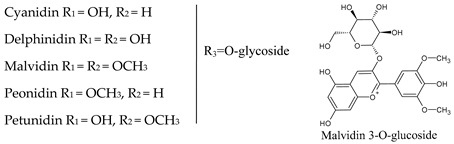
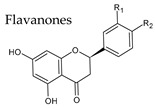	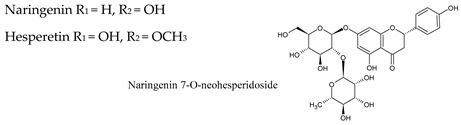
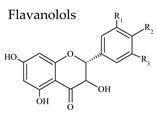	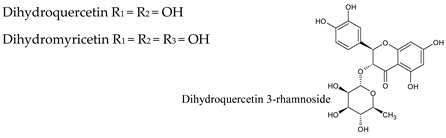
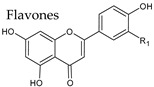	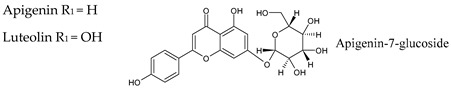
**Non-flavonoids**	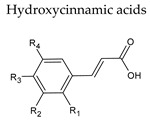	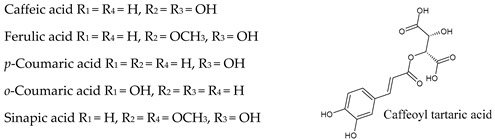
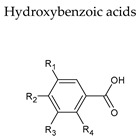	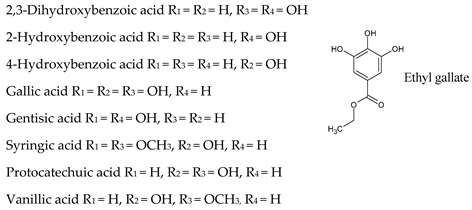
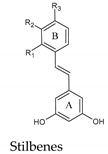	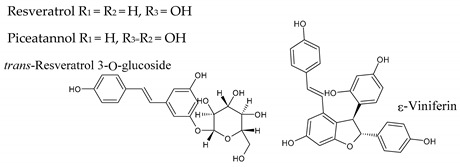

**Table 2 molecules-25-04105-t002:** Range of total phenolic content in red and white wines in a representative set of studies.

	Red Wine			White Wine			
	Total Phenolic Content ^a^			Total Phenolic Content ^a^			
	Range			Range			
	Min.	Max.	*n*	Min.	Max.	*n*	Reference
	1615	4177	7	216	854	7	[16]
	1313	2389	16	89	407	17	[17]
	2193	3183	6	292	402	4	[18]
	622	3200	20	-	-	-	[19]
	1724	1936	5	282	434	5	[15]
	2340	3730	23	-	-	-	[20]
	1460	3380	39	210	390	47	[21]
	2082	3184	3	213	277	5	[22]
	1402	3180	24	189	425	11	[14]
	3200	5900	8	-	-	-	[23]
	1788	3070	4	55	370	20	[24]
	1012	3264	11	-	-	-	[25]
	554	2669	2	167	347	3	[26]
	1181	3589	23	-	-	-	[27]
	-	-	-	291	2103	14	[28]
	860	2710	8	-	-	-	[29]
	1602	1968	7	-	-	-	[30]
	1837	3467	6	-	-	-	[31]
	896	6319	38	77	83	2	[32]
**Mean ± SD**	1538 ± 664	3406 ± 1139		189 ± 85	554 ± 545		
**Median (Q25–Q75)**	1531 (983–1898)	3192 (2700–3624)		210 (89–282)	402 (347–434)		

^a^ Data are expressed as gallic acid equivalents (GAE) in mg/L.

**Table 3 molecules-25-04105-t003:** (Poly)phenol contents in red and white wines.

		Red Wine						White Wine					
		Phenol Explorer ^a^			USDA ^b^			Phenol Explorer ^a^			USDA ^b^		
		mean	min	max	mean	min	max	mean	min	max	mean	min	max
**Flavonoids**	**Main representatives**												
Anthocyanins	Cyanidin ^c^	2.9	0.6	11.9	1.9	0.0	45.0	-	-	-	-	-	-
	Delphinidin ^c^	16.6	2.4	40.1	20.1	0.2	57.1	-	-	-	-	-	-
	Malvidin ^c^	156	12.4	541	138	0.0	536	0.4	0.0	3.5	0.6	0.0	2.4
	Peodinin ^c^	18.1	2.5	80.9	12.5	0.2	50.3	-	-	-	-	-	-
	Petunidin ^c^	23.6	3.4	61.8	19.8	0.2	56.6	-	-	-	-	-	-
Total		217	21.3	736	193	0.6	745	0.4	0.0	3.5	0.6	0.0	2.4
Dihydroflavonols	Dihydromyricetin 3-*O*-rhamnoside	44.7	44.7	44.7	-	-	-	3.0	3.0	3.0	-	-	-
Total		54.4	45.8	59.8	-	-	-	5.7	3.7	15.9	-	-	-
Flavanols	(+)-Catechin	68.1	13.8	390	71.4	0.0	390	10.8	0.0	46.0	7.7	0.0	58.0
	(-)-Epicatechin	37.8	0.0	165	37.9	0.0	165	9.5	0.0	60.0	5.5	0.5	60.0
	Proanthocyanidins	355	99.7	560	296	63.1	1354	0.2	0.0	1.5	3.9	0.6	7.3
Total		470	114	1131	407	63.1	1917	20.8	0.0	109	17.1	1.1	125
Flavanones	Naringenin ^c^	8.0	7.3	8.8	17.7	10.3	25.1	2.3	1.7	2.9	3.8	0.0	7.7
Total		8.5	7.8	9.4	24.0	13.0	35.0	2.3	1.7	2.9	7.8	3.2	12.5
Flavones	Apigenin	-	-	-	1.3	0.0	4.7	-	-	-	-	-	-
Total		-	-	-	1.7	0.0	8.7	-	-	-	-	-	-
Flavonols	Isorhamnetin ^c^	5.9	1.7	11.6	0.2	0.0	1.6	0.0	0.0	0.0	0.0	0.0	0.2
	Kaempferol ^c^	10.2	5.7	14.4	0.9	0.0	13.7	0.2	0.0	2.6	0.1	0.0	2.7
	Myricetin ^c^	8.3	0.0	17.9	4.2	0.0	17.9	0.0	0.0	0.0	0.1	0.0	1.0
	Quercetin ^c^	44.2	12.3	110	10.4	0.0	33.6	4.6	1.3	20.8	0.4	0.0	8.4
Total		68.6	19.7	154	15.7	0.0	66.8	4.8	1.3	23.4	0.5	0.0	9.4
**Non-flavonoids**													
Hydroxybenzoic acids	Gallic	35.9	0.0	126	NA	NA	NA	2.2	0.0	11.0	NA	NA	NA
	Gentisic	4.6	0.0	8.0	NA	NA	NA	18.2	0.0	20.0	NA	NA	NA
	Protocatechuic	1.7	0.0	9.6	NA	NA	NA	3.3	0.1	13.0	NA	NA	NA
	Syringic	2.7	0.0	23.3	NA	NA	NA	0.5	0.0	0.2	NA	NA	NA
	Vanillic	3.2	0.0	7.5	NA	NA	NA	0.4	0.1	1.2	NA	NA	NA
Total		70.1	13.7	221	NA	NA	NA	24.8	0.4	46.8	NA	NA	NA
Hydroxycinnamic acids	Caffeic	18.8	0.0	77.0	NA	NA	NA	2.4	0.0	7.0	NA	NA	NA
	Caftaric	33.5	1.4	179	NA	NA	NA	21.5	21.4	22.0	NA	NA	NA
	Ferulic	0.8	0.0	10.4	NA	NA	NA	0.9	0.3	2.1	NA	NA	NA
	(*o*- and *p*-) Coumaric	5.8	0.2	40.4	NA	NA	NA	1.8	0.0	5.6	NA	NA	NA
	Sinapic	0.7	0.0	5.4	NA	NA	NA	0.6	0.0	2.8	NA	NA	NA
Total		100	14.9	378	NA	NA	NA	28.2	21.7	42.4	NA	NA	NA
Stilbenes	Resveratrol ^d^	5.8	0.0	61.5	NA	NA	NA	0.9	0.0	3.2	NA	NA	NA
	Resveratrol 3-O-glucoside ^d^	12.5	0.0	88.0	NA	NA	NA	5.0	0.4	13.1	NA	NA	NA
	Piceatannol ^c^	15.3	6.3	38.8	NA	NA	NA	4.6	1.4	8.0	NA	NA	NA
	Viniferins (δ-. ε-)	7.9	0.1	26.7	NA	NA	NA	0.6	0.0	0.1	NA	NA	NA
	Pallidol	2.0	0.0	2.5	NA	NA	NA	0.7	0.0	0.3	NA	NA	NA
Total		43.5	6.4	218	NA	NA	NA	10.6	1.4	24.7	NA	NA	NA
Other polyphenols	Hydroxybenzaldehydes	7.1	0.0	45.6	NA	NA	NA	4.1	2.4	5.8	NA	NA	NA
	Tyrosols	36.5	6.4	54.3	NA	NA	NA	4.2	2.7	5.7	NA	NA	NA
Total		43.6	6.4	99.9	NA	NA	NA	8.3	5.1	11.5	NA	NA	NA

^a^ Data are expressed as milligrams of gallic acid equivalent per litter (mg/GAE/L). ^b^ Data are expressed as milligrams of gallic acid equivalent per kilogram (mg/GAE/Kg). ^c^ Plus glucoside derivatives.^d^ includes both cis and trans conformations. NA, data not available in the database.

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
