# Peer review of "Wine’s Phenolic Compounds and Health: A Pythagorean View"

_molecules, 2020, doi:10.3390/molecules25184105_

Round 1

Reviewer 1 Report

This is a very interesting paper on a hot topic. i have some suggestions:

  1. Please add references in the text and discuss on the hepatic microsomal ethanol oxidizing system (MEOS), which converts ethanol to acetaldehyde and produces ROS as a by-product via cyp 2E1. Are polyphenols of red wine protective in the liver where ROS is formed?
  2. Polyphenols are wrongly considered as healthy and protective in many diseases, please reference in the text and discuss in more detail many more diseases as well as aging and longevity in Okinawa.  Issue remains controversial. 

Author Response

This is a very interesting paper on a hot topic. I have some suggestions:

  1. Please add references in the text and discuss on the hepatic microsomal ethanol oxidizing system (MEOS), which converts ethanol to acetaldehyde and produces ROS as a by-product via cyp 2E1. Are polyphenols of red wine protective in the liver where ROS is formed?

Interesting point that lack solid evidence. We added a paragraph on this and we believe it would be worth exploring it.

  1. Polyphenols are wrongly considered as healthy and protective in many diseases, please reference in the text and discuss in more detail many more diseases as well as aging and longevity in Okinawa.  Issue remains controversial.

We respectfully disagree. Even though there is too much emphasis on polyphenols, epidemiological studies are consistent: high polyphenol intake affords better health. The mechanisms behind this effect are not entirely clear and we wasted many years touting the antioxidant properties of polyphenols. Okinawans have a very peculiar diet, rich in carbohydrates and low in protein and calories. In addition, they eat a lot of omega 3 fatty acids and polyphenols (via roots and algae). It is very difficult to single out the contribution of any of these components.

Reviewer 2 Report

The review describes the benefits of wine consumption and its influence on health. It provides a comprehensive compilation of data on grape components.
The contributions of grapes and wine have been widely reported in original and review articles as well as in books. This rationale has led to the concept of and several literatures on superfoods. It is not clear what contribution this manuscript is making to already existing knowledge because much similar work was previously published in this matter.
Therefore where is the novelty?

Author Response

Referee 2

The review describes the benefits of wine consumption and its influence on health. It provides a comprehensive compilation of data on grape components.
The contributions of grapes and wine have been widely reported in original and review articles as well as in books. This rationale has led to the concept of and several literatures on superfoods. It is not clear what contribution this manuscript is making to already existing knowledge because much similar work was previously published in this matter.
Therefore where is the novelty?

This is an invited review, part of the proceedings of a meeting. There are no such papers in the literature (unless the referee can prove otherwise) because much literature is over-emphasizing the roles of wine polyphenols in health. Luckily, the other two reviewers appreciated our effort.

Reviewer 3 Report

The review paper entitled Wine's minor components and health: a Pythagorean view was very interesting to read, sometimes provocative but with always based on the best evidence available. Therefore the paper should be accepted for publication. I would suggest to change the title as the paper only reviews phenolic compounds from wine, and there are a lot of other minor components in wines. Perhaps Wine's phenolic compounds and health: a
Pythagorean view should describe better the content of the paper.

Other minor suggestions/corrections

Line 76 – vinification

Table 1 –

Flavan-3-ols and not Flavan-3-oles

Missing R5 in Flavan-3-ols

Structure of Flavanonols should be made identical to the others with the R attached to the benzene ring

Protocatechuic R3=OH

Syringic R2=OH; R1=R3=OCH3

2,3-dihydroxibenzoic acid R4=OH R3=OH

Line 99 alkalization???

Points 4, 5, 6 … should be sub-points of 3

Points 5, 6, 7 are in fact subpoints of 4

Line 121 Flavan-3-ols

Line 149 better: lack the double bond between carbons 2 and 3 in the C-ring of the flavonoid skeleton

Line 180-181 - which consists in a 181 conjugation of caffeic and tartaric acid: Better is composed by caffeic acid esterified with tartaric acid

Line 199 while ring B is substituted with …

Line 199 Hydroxyl groups

Line 199 Methoxyl groups

Table 4 – This is a very important table to support the discussion of the manuscript and backup the idea defended by the authors, nevertheless is very difficult to read and less intuitive. Perhaps the authors could rearrange this table and put the results of the interventions not as columns but as lines below each study. Just a suggestion.

Author Response

Referee 3

The review paper entitled Wine's minor components and health: a Pythagorean view was very interesting to read, sometimes provocative but with always based on the best evidence available. Therefore the paper should be accepted for publication. I would suggest to change the title as the paper only reviews phenolic compounds from wine, and there are a lot of other minor components in wines. Perhaps Wine's phenolic compounds and health: a Pythagorean view should describe better the content of the paper.

Good idea. We changed the title accordingly.

Other minor suggestions/corrections

Line 76 – vinification

Indeed….

Table 1 –

Flavan-3-ols and not Flavan-3-oles

Done

Missing R5 in Flavan-3-ols

Done

Structure of Flavanonols should be made identical to the others with the R attached to the benzene ring

Done

Protocatechuic R3=OH

Done

Syringic R2=OH; R1=R3=OCH3

Done

2,3-dihydroxibenzoic acid R4=OH R3=OH

Done

Line 99 alkalization???

Indeed, it is alkalinization, now corrected.

Points 4, 5, 6 … should be sub-points of 3

Points 5, 6, 7 are in fact subpoints of 4

This was done by MDPI. We will make sure it’s corrected in the final version.

Line 121 Flavan-3-ols

Corrected

Line 149 better: lack the double bond between carbons 2 and 3 in the C-ring of the flavonoid skeleton

Corrected

Line 180-181 - which consists in a 181 conjugation of caffeic and tartaric acid: Better is composed by caffeic acid esterified with tartaric acid

Corrected

Line 199 while ring B is substituted with …

Done

Line 199 Hydroxyl groups

Done

Line 199 Methoxyl groups

Done

Table 4 – This is a very important table to support the discussion of the manuscript and backup the idea defended by the authors, nevertheless is very difficult to read and less intuitive. Perhaps the authors could rearrange this table and put the results of the interventions not as columns but as lines below each study. Just a suggestion

Great suggestion, which we followed. Please note that, while this paper was under review, we searched the literature for additional acute studies. We now have another Table (5) that follows the same scheme you suggested.

Round 2

Reviewer 1 Report

My comments were fairly considered.

Reviewer 2 Report

Present paper after revision is suitable for accepting for publishing in Molecules journal.